# Deciphering the functional diversity of DNA-binding transcription factors in Bacteria and Archaea organisms

Emanuel Flores-Bautista[1], Rafael Hernandez-Guerrero[1], Alejandro Huerta-Saquero[2], Silvia Tenorio-Salgado[3], Nancy Rivera-Gomez[4], Alba Romero[5], Jose Antonio Ibarra[6], Ernesto Perez-Rueda[1,7]*

1 Instituto de Investigaciones en Matemáticas Aplicadas y en Sistemas, Universidad Nacional Autónoma de México, Unidad Académica Yucatán, Mérida, Yucatán, México, 2 Centro de Nanociencias y Nanotecnología, Universidad Nacional Autónoma de México, Ensenada, Baja California, México, 3 Tecnológico Nacional de México, Instituto Tecnológico de Mérida, Mérida, Yucatán, México, 4 Instituto Nacional de Salud Pública, Cuernavaca, Morelos, México, 5 Microbiota Host Interactions and Clostridia Research Group, Universidad Nacional Andrés Bello, Santiago de Chile, Chile, 6 Laboratorio de Genética Microbiana, Departamento de Microbiología, Escuela Nacional de Ciencias Biológicas, Instituto Politécnico Nacional, Ciudad de México, México, 7 Centro de Genómica y Bioinformática, Facultad de Ciencias, Universidad Mayor, Santiago, Chile

* ernesto.perez@iimas.unam.mx

**Data Availability Statement:** All relevant data are within the paper and its Supporting Information files.

## Abstract

DNA-binding Transcription Factors (TFs) play a central role in regulation of gene expression in prokaryotic organisms, and similarities at the sequence level have been reported. These proteins are predicted with different abundances as a consequence of genome size, where small organisms contain a low proportion of TFs and large genomes contain a high proportion of TFs. In this work, we analyzed a collection of 668 experimentally validated TFs across 30 different species from diverse taxonomical classes, including *Escherichia coli* K-12, *Bacillus subtilis* 168, *Corynebacterium glutamicum*, and *Streptomyces coelicolor*, among others. This collection of TFs, together with 111 hidden Markov model profiles associated with DNA-binding TFs collected from diverse databases such as PFAM and DBD, was used to identify the repertoire of proteins putatively devoted to gene regulation in 1321 representative genomes of Archaea and Bacteria. The predicted regulatory proteins were posteriorly analyzed in terms of their genomic context, allowing the prediction of functions for TFs and their neighbor genes, such as genes involved in virulence, enzymatic functions, phosphorylation mechanisms, and antibiotic resistance. The functional analysis associated with PFAM groups showed diverse functional categories were significantly enriched in the collection of TFs and the proteins encoded by the neighbor genes, in particular, small-molecule binding and amino acid transmembrane transporter activities associated with the LysR family and proteins devoted to cellular aromatic compound metabolic processes or responses to drugs, stress, or abiotic stimuli in the MarR family. We consider that with the increasing data derived from new technologies, novel TFs can be identified and help improve the predictions for this class of proteins in complete genomes. The complete collection of experimentally characterized and predicted TFs is available at *http://web.pcyt.unam.mx/EntrafDB/*.

**Funding:** This study was supported by the Dirección General de Asuntos del Personal Académico-Universidad Nacional Autónoma de México (IN-209620).

**Competing interests:** The authors have declared that no competing interests exist.

## Introduction

In recent years, the great number of organisms completely sequenced has shown that the genomes are the products of diverse evolutionary events, such as gene expansion, gene loss, and lateral gene transfer [1, 2]. However, it has been already described that all the organisms share common principles of gene regulation across large phylogenetic distances [3–5]. In this regard, gene repertoire and regulatory mechanisms are important for appropriate responses and adaptations to environmental challenges [6, 7]. Gene expression is modulated predominantly at the transcription initiation level through DNA-binding Transcription Factors (TFs), which provide organisms with the ability to express different genes in response to internal and external stimuli. TFs repress or activate gene expression via blocking or allowing access of the RNA polymerase to the promoter, depending on the operator context and ligand-binding status [8, 9]. Due to the crucial role of TFs in coordinating gene expression, this class of proteins has been evaluated in complete bacterial and archaeal genomes based on sequence comparisons and utilizing as seeds those proteins with experimental evidence, such as those in *Escherichia coli* K-12 [10], *Bacillus subtilis* 168 [4], and *Staphylococcus aureus* [5], among others. In this context, diverse databases describing TFs as a consequence of predictive methods have been proposed, such as the DBD database [11], Archaea TF database [12], and P2TF database [13]. However, to the best of our knowledge, a systematic comparison between the sequence predictions and experimental evidences has not been sufficiently described. In this study, we identified proteins devoted to gene regulation in 5321 genomes of Archaea and Bacteria, based on PFAM annotations and a set of well-annotated TFs. In addition, to determine the probable functions regulated by these TFs, we evaluated the genomic context for the most abundant families identified in the dataset. We consider that with the increasing data derived from the new technologies, novel TFs can be identified and might be able to help improve prediction for this class of proteins in complete genomes.

## Materials and methods

### Collection of well-known TFs

In order to obtain a validated dataset associated with TFs, first we searched the Uniprot database (version October 10, 2018) to retrieve all probable TFs by using the following keywords (and their combinations): Transcription; Gene regulation; DNA-binding; Repression; Activation; Repressor; Activator; Regulator; Bacteria; Archaea. As a result, 967 protein records were identified as potential TFs. From these records and to ensure the quality of the obtained data, we manually verified these records. In addition, we performed literature look-up and BLAST searches against a nonredundant (NR) database at NCBI, with E-values less than $10^{-3}$ to identify proteins with not known regulatory roles, such as peptidases, lipolytic enzymes, integrases, and transporters. Therefore, for each collected TF, its ID, protein sequence, and functional description were considered, along with the resources for its corresponding IDs in available databases. Furthermore, we scanned diverse well-annotated databases in order to retrieve diverse TFs in prokaryotes, including the RegulonDB [14], DBTBS [15], and DBD databases [11], along with results from literature searches, to identify recent, newly reported TFs. From these searches, we identified a total of 668 proteins as TFs. Given the different nomenclature used for these various databases and curations to assign information from experimental evidence, we normalized the nomenclature categories as follows: Mutagenesis; Footprint; Crystallographic information; Transcriptional fusions; Protein structural information (from PDB); Binding of purified proteins (BPP); and Binding of cellular extracts (BCE), among others (see Table 1).

**Table 1. Experimental evidences considered in this work.**

| Evidence | Quality | Description | Total no. of records |
|---|---|---|---|
| PDB | Strong | Protein Data Bank; protein structure-based inferences on regulation, based on observations of protein structures | 245 |
| BPP | Strong | Binding of Purified Proteins; footprinting assay results (DNase I, DMS, EMSA, etc.) | 292 |
| SM | Strong | Site Mutation; e.g., a cis-mutation in the DNA sequence of a TF binding site interferes with the regulatory function | 165 |
| IDA | Strong | Inferred from Direct Assay | 6 |
| APPH | Strong | Assay of Protein Purified to Homogeneity; e.g., in vitro transcription assay | 6 |
| qRT-PCR | Strong | Quantitative reverse transcription-PCR; mRNA expression levels of a regulated element are measured and compared between wild-type and trans-element mutants (knockout, overexpression, etc.) | 11 |
| OHR | Strong | One-Hybrid Reporter system; physical binding of the regulator to its regulated promoter | 25 |
| ChIP-seq | Strong | Chromatin immunoprecipitation sequencing; physical binding of the regulator to its regulated promoter | 119 |
| PRM | Strong | PRiMer extension analysis; transcription initiation mapping (in combination with transcript concentration measurements) to compare mutant vs wild-type expression levels | 2 |
| BCE | Weak | Binding of Cellular Extracts; e.g., gel shift analysis | 108 |
| GEA | Weak | Gene Expression Analysis; e.g., expression levels of a LacZ-regulated promoter fusion compared between wild type and trans-element mutants (knockout, overexpression, etc.) | 257 |
| AS | Weak | Author Statement; traceable author statement to experimental support | 21 |
| IEP | Weak | Inferred from Expression Pattern | 6 |
| IMP | Weak | Inferred from Mutant Phenotype; e.g., a mutated TF has a visible cell phenotype, and it is inferred that the regulator might regulate the genes responsible for the phenotype | 26 |
| IGI | Weak | Inferred from Genetic Interaction; e.g., in vitro titration assay | 15 |
| RBM | Weak | Reaction Blocked in Mutant; gene inactivation | 4 |
| MIC | Weak | MICroarrays; mRNA levels of a regulated element compared between wild type and trans-element mutants (knockout, overexpression, etc.), performed by using microarray (or macroarray) experiments | 38 |

## Bacterial and archaeal genomes analyzed

To analyze bacterial and archaeal organisms, 5321 genomes from the NCBI Refseq genome database [16] were downloaded, and a web-based tool with a Genome Similarity Score (GCCa) of $\geq 0.95$ [17] was considered to select nonredundant genomes, accounting for a set of 1321 representative genomes from 106 archaeal species and 1245 bacterial species.

## Identification of DNA-binding domains (DBDs) associated with TFs

A total of 16,712 hidden Markov models (HMMs) retrieved from the PFAM database version 17.0 were used to scan the 668 well-known TFs and the 5321 genomes by using the program pfam_scan.pl with an E-value $\leq 10^{-3}$, with the option of clan_overlap activated (to show over-lapping hits within clan member families). In addition, 111 PFAMs associated with DNA-binding TFs were retrieved from regulatory proteins deposited in diverse databases, such as the DBD, RegulonDB, and DBTB databases and those identified by manual curation. Therefore, proteins associated with the 111 PFAMs were considered TFs.

## Identification of orthologous proteins

To identify orthologous proteins, the program Proteinortho [18] was applied. We used this program because it implements an extended version of the reciprocal best heuristic alignment [18]; reduces the amount of memory required for orthology analysis, when compared to OrthoMCL and Multi-Paranoid, and the performance is comparable with OrthoMCL [19]. To this end, we computed the orthologous proteins in the complete set of all 668 proteins against the 5321 bacterial and archaeal genomes, employing an E-value $\leq 10^{-5}$, with a coverage of

≥70%, considered as significant to identify orthologous proteins against the collection of well-known proteins.

## Identification of virulence factors on TFs and neighbor genes

To identify proteins probably related to virulence, a set of 3224 proteins retrieved from the Virulence Factor Database (VFDB) [20] were compared against the proteins of the 5321 bacterial and archaeal genomes, with an E-value $\leq 10^{-5}$ and a coverage of ≥70%, using the program Proteinortho. This dataset considers only proteins associated with experimentally verified virulence factors and includes bacterial toxins, cell surface proteins that mediate bacterial attachment, cell surface carbohydrates and proteins that protect a bacterium, and hydrolytic enzymes that may contribute to the bacterial pathogenicity [20].

## Genomic context

In order to analyze the genomic context of TFs, we downloaded the predictions of Transcription Units (TUs) or operons from the website https://microbiome.wordpress.com/research/predicting-transcription-units/ [21]. In brief, the predictions are based on the transcription direction and their intergenic distance, i.e., short distances are frequent in genes belonging to the same TU, considering that predicted operons contain a higher proportion of genes with related phylogenetic profiles and conservation of adjacency than predicted borders of TUs. This method was used because of its high performance to identify correctly operons for *E. coli* and *B. subtilis* [22]. In addition, divergently oriented genes relative to the TF and their intergenic distances were computed.

## Identification of enzymatic activities

To determine if TFs and their neighbor genes are associated with enzymatic activities, the Catalytic Families (CatFam version 2.0) program was used to scan the complete set of proteins associated with the 1321 representative bacterial and archaeal genomes, using default conditions. We used, CatFam because it generates 8880 sequence profiles through a stepwise procedure that carefully controls profile quality and employs nonenzymes as negative samples to establish profile-specific thresholds associated with a predefined nominal false-positive rate of predictions; i.e. it predicts enzymes with 98.6% precision and 95.0% recall [23].

## PFAM functional enrichment

To evaluate the enriched PFAM associated with the top 10 TFs and their neighbors, their structural domains were determined by considering the PFAM assignments. A one-tailed Fisher's exact test was used to perform enrichment analysis, because it is related to the hypergeometric probability and can be used to calculate the significance (p-value) of the overlap between two independent datasets. We set statistical significance at a p-value of 0.05. In addition, the false discovery rate (FDR) of the tests was calculated to account for type I errors. Corrections for multiple testing were performed using the Benjamini-Hochberg step-up FDR-controlling procedure to calculate adjusted p-values. Finally, we linked the PFAM identified for the functional annotations provided by using *pfam2go*, which considers mapping between PFAMs and *G*ene Ontology (GO) terms.

# Results and discussion

## Collection of TFs with experimental evidences

In order to have a collection of DNA-binding TFs well-annotated for use as a gold standard dataset to guide the design and predictions of new regulators of bacteria and archaea, a total of

668 proteins with experimental evidence were collected. To this end, we defined a TF as DNA-binding protein needed to activate or repress the transcription of a gene, but are themselves neither part of the RNA polymerase (RNAP) core nor of the holoenzyme [24]. Therefore, sigma factors were not considered as TFs in this study. A total of 842 references and diverse databases (Uniprot, DBD, DBTBS, and RegulonDB, among others) were manually inspected. The complete collection of TFs was identified in 81 different species, including *E. coli*, *Mycobacterium tuberculosis*, *B. subtilis*, *Corynebacterium glutamicum*, *S. aureus*, *Streptomyces coelicolor*, and many others; thus, we considered organisms from diverse taxonomic divisions, from Bacteria to Archaea and from Proteobacteria to Euryarchaeaota (Table 1).

The experimental sources associated with the 668 TFs are diverse, such as site mutations (site-directed mutagenesis in the DNA-binding site recognized by the TF), genetic interactions (*in vitro* titration assay results), physical binding of the regulator to the regulated promoter (demonstrated by footprinting assays), expression levels of LacZ-regulated promoter fusions measured and compared between wild-type and trans-element mutants (knockout, overexpression, etc.), and mRNA levels of regulated element measured and compared between wild-type and trans-element mutants (knockout, overexpression, etc.), performed by using microarray (or macroarray) experiments; proteomic studies [regulated gene product concentrations measured and compared between wild-type and trans-element mutants (knockout, overexpression, etc.)] using proteomics techniques (Table 1). The strengths of all these experimental evidence types were assigned using the following criteria: (i) strong evidence, which suggests that the TF is clearly associated with the binding site; (ii) weak evidence, for a validated TF for which there is some low-throughput experimental evidence to verify its function. The collection of TFs can be accessed at *http://web.pcyt.unam.mx/EntrafDB/*.

In terms of structural domains, 41.77% of total proteins are monodomain according to PFAM, whereas 50.22% are two-domain proteins, 6.03% have three domains, 0.9% have four domains, 0.2% have five domains, and 0.7% have six or seven domains. In general, it has been accepted that proteins can be classified into families. In this work, we considered PFAM assignments to identify proteins of similar evolutionary groups. According to PFAM, the most abundant family identified in the collection corresponds to the TetR/AcrR family, which represents 9.8% of the total collection, followed by the family Trans_reg_C or OmpR/PhoB (7.9%) (Fig 1). The abundance of studies on these proteins could be associated with their functions, as they play fundamental roles in resistance to multiple compounds, such as antibiotics, e.g., CecR of *Escherichia coli*, which is involved in the regulation of the *cecR-ybhGFSR* operon and the *rhlE* gene, which together are involved in the control of sensitivity to cefoperazone and chloramphenicol [25]. In contrast, 22% of the TF collection are classified into a wide diversity of families with small numbers of members, such as the SarA family, which includes the cytoplasmic regulator; induction of exoproteins and repression of spa for virulence in animal models of biofilm infection [26], or NrdR of *E. coli*, which is associated with the family NrdR [27] which plays a fundamental role in the regulation of the ribonucleotide reductases (RNRs) that catalyze the conversion of ribonucleotides to deoxyribonucleotides and are essential for de novo DNA synthesis and repair [28].

Finally, the experimentally characterized TFs are mainly associated with repression of genes (29.4%), followed by TFs that can activate and/or repress gene expression, 23.9%, and a low proportion of activators (18.1%). Proteins with functions not clearly defined represents the 28% of the collection. This finding suggests that repression is the most abundant regulatory mechanisms, as it has been described in the *E. coli* K12 [10] and *B. subtilis* 168 [4], and it is consistent with hypothesis that most promoters are repressed in bacteria, mainly by steric hindrance, where the repressor-binding site overlaps core promoter elements and blocks recognition of the promoter by the RNAP holoenzyme [29].

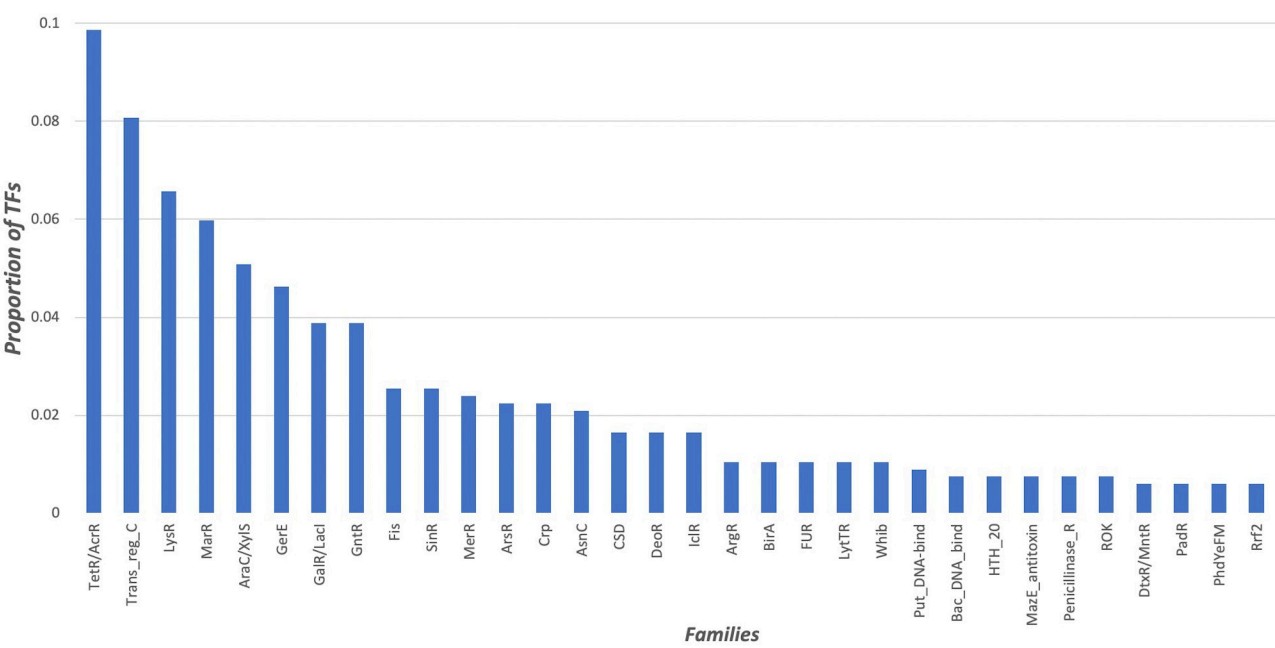

**Fig 1. Distribution of TF families.** In x axis is indicated the 32 families identified in this work represent 78% of the total proteins identified in this version. The abundance of TFs is shown on the y axis.

## Predictions of TFs with PFAM assignments and orthology analysis

In order to identify the repertoire of DNA-binding TFs, two approaches were employed. The first approach considers an exhaustive scanning with the PFAM assignments on the prokaryotic genomes; from there, 111 PFAM domains associated with DNA-binding TFs were retrieved. The second approach considers an ortholog analysis and uses the Proteinortho program [18] and is based on a collection of 668 well-known TFs. Based on this approach, the Pearson correlation coefficient was 0.88 (p-value $< 2.2e216$), showing a strong positive correlation between TFs and genome size (measured by ORFs number). In addition, the power fit function exponent (1.52) was within the range reported in other studies for protein families classified as regulators [30]. Fig 2.

Previous studies to predict putative TFs in complete genomes depend on similarity search strategies, under the hypothesis of functional and evolutionary relationships between homologous proteins. In general, BLASTp, RPS-BLAST, and HMM searches considering the *E. coli* dataset as reference, have been implemented to predict these proteins in bacterial and archaeal genomes, such as the Archaea TF [12], P2TF [13], or DBD [11]. These approaches allow to infer that around 10% of the gene products are devoted to regulate gene expression, with exceptions in archaeal genomes, where an apparent deficit of TFs has been suggested [31]. In this work, we consider two main sources of information to predict TFs in all the genomes, as it was described in methods; the PFAM assignments associated to DBDs and retrieved from diverse databases and the set of well-known TFs.

To evaluate the accuracy of the prediction process, we compared our predictions against the repertoire described in bacterial models. For instance, in *E. coli* K12 we predicted a total of 336 TFs. From these, 103 proteins were exclusively predicted by HMM searches, 203 by PFAM models and by sequence comparisons against the well-known dataset, whereas 30 proteins were only predicted by sequence comparisons. In *B. subtilis* 168, we predicted 286 TFs, 118 by HMM searches, 123 by both approaches, and 43 by sequence comparisons. Thus, 8.9% and

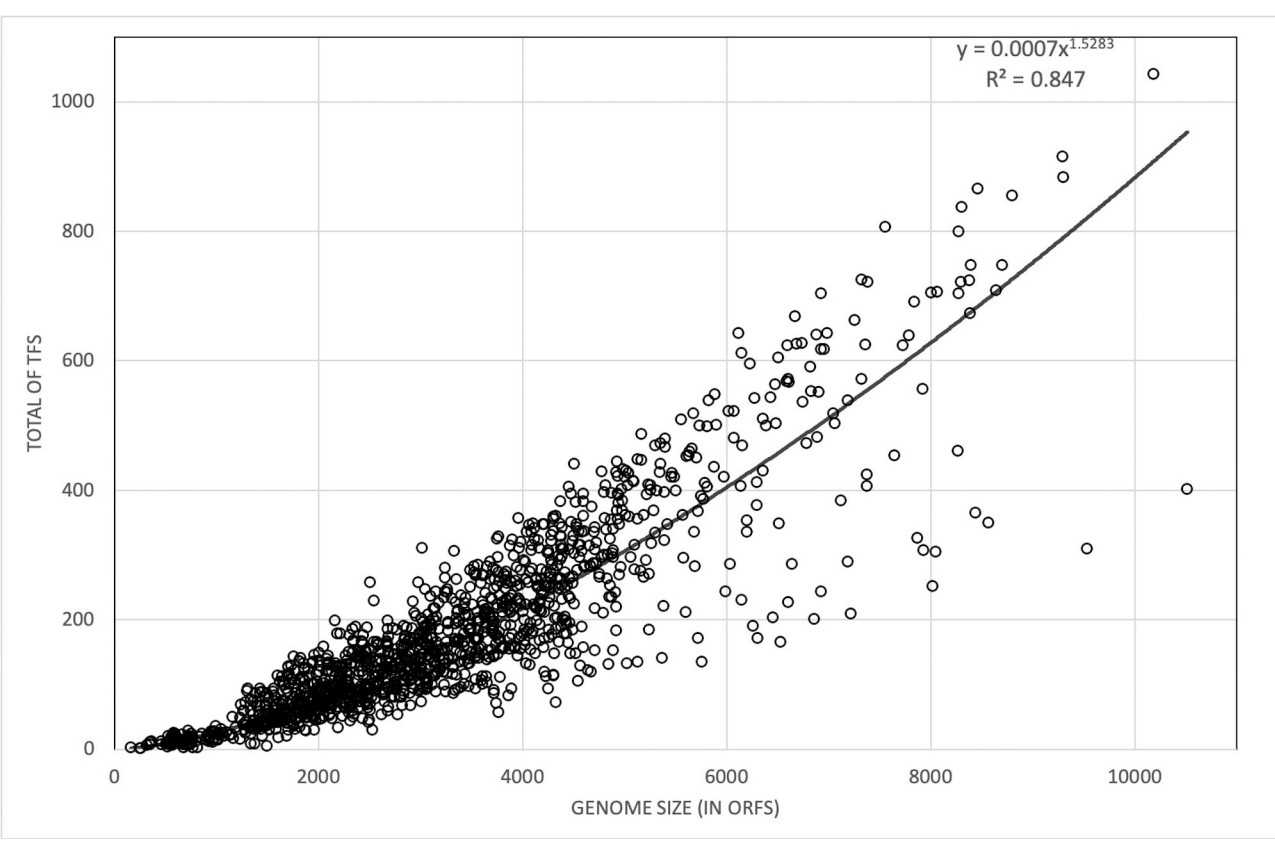

**Fig 2. Abundance of TFs in bacteria and archaeal genomes as a function of genome size.** The abundance of TFs for each genome is shown on the y axis. Each dot corresponds to one genome. The power fit function (black line) and $R^2$ are indicated, y = 0.0007 x1.5283.

15.0% of the predictions of TFs in *E. coli* and *B. subtilis*, respectively, were only obtained by sequence comparisons, suggesting that all predictions are increasing when our dataset of experimentally described TFs is considered.

We consider that our set of known and predicted TFs in both genomes is similar to previous works [4, 32], and that we are close to the total number of TFs in these bacteria. Therefore, the existence of alternative regulatory mechanisms, such as riboswitches, DNA-curvature or attenuation could influence gene expression where there is no evidence of regulation mediated by TFs, as occurs in organisms where less proportion of TFs was predicted; such as occurs in the delta proteobacterium *Sorangium cellulosum* genome (GCF_000418325) with 10514 ORFs and 402 predicted TFs, i.e. 3.8% of its gene products. In particular, Hang et al [33] identified a significant abundance of eukaryote-like protein kinases (508 proteins), 169 sigma factors, 6 anti-sigma factor proteins, and 347 sigma factor-related proteins; suggesting that the regulation of gene products are mainly related to kinases and sigma factors than TFs [33].

Therefore, to evaluate if the relation between TFs and genome size is uniform for all the bacterial and archaeal divisions, a subset of genomes from nine divisions, with a widespread distribution and that represented 81% of the total of NR genomes analyzed, was studied. In this regard, the TFs fit a linear function in *Actinobacteria*, *Firmicute*, and delta *Proteobacteria* from Bacteria and in *Euryarchaeota* from Archaea cellular divisions. In contrast, the repertoire of TFs fits a power function in gamma, alpha, and beta *Proteobacteria*, the CFB (*Cytophaga*, *Fusobacterium*, and *Bacteroides*) group, and cyanobacteria (S1 Fig). This result suggests that the relationship between TFs and genome size fit a linear and power function in the organisms

analyzed, indicating that bacterial and archaeal divisions follow diverse strategies to modulate repertoires of regulatory proteins. The complete collection of predictions is available at *http://web.pcyt.unam.mx/EntrafDB/*, section "predictions".

## TFs with noncanonical DBDs

Previous works have described that the most abundant DNA-binding structure described so far in bacteria and archaea is the helix-turn-helix (H-T-H). Indeed, preliminary calculations suggest that around 80% of DNA-binding TFs contain a H-T-H [34]; whereas a heterogeneous collection of proteins with diverse structural domains represents the other 20%, such as the ribbon-helix, the LytR, or the cold shock domain [35]. Although this distribution remains with no significant changes, in recent years various analyses have found that the diversity of structural domains is larger than expected, i.e., the emergence of new DNA-binding motifs is documented in the literature, opening the opportunity to identify novel structures of TFs. In the next sections, we illustrate some of the most significant and noncanonical DNA-binding structures that were identified in the collection of experimentally characterized TFs and that were used to predict regulatory proteins in complete bacterial and archaeal genomes.

## LytTr DBD

The LytTr DBD (PFAM ID PF04397) has been found in diverse proteins from the two-component signal transduction system, such as AlgR of *Pseudomonas aeruginosa*, which is involved in the regulation of alginate biosynthesis and in the pathogenesis of cystic fibrosis. Indeed, most TFs described with this structural domain have been associated with the biosynthesis of extracellular polysaccharides, fimbriation, expression of exoproteins including toxins, and quorum sensing. The LytTR domain consists of three α-helices and four β-strands with a potential additional short β-strand between the first and second α-helices. Remarkably, the conserved F[FYVL][RQ][CIV] motif forms an additional β-strand and a loop (or turn) structure. In addition, this region has no apparent similarity to the turn regions of the classical H-T-H [36]. Based on the identification of these proteins in the complete prokaryotic genomes, it was found that although the LytTr has a wide distribution along the organisms analyzed (721 out 1351 representative genomes; 53.3%), they are present in low proportions (one protein per genome).

## ArfGap/RecO-like zinc finger domain (PFAM ID PF05443)

The TF Ros identified in *Agrobacterium tumefaciens* (Uniprot ID Q04152) contains a Cys(2)His(2) zinc-finger domain that is part of a significantly larger zinc-binding globular domain. This domain possesses a novel protein fold very different from the classical fold reported for eukaryotic zinc fingers. Mapping of the amino acids necessary for DNA binding onto the Ros structure revealed the protein surface involved in the DNA recognition mechanism of this new zinc-binding protein domain [37]. This family is sparsely distributed and does not exhibit a wide distribution across the bacterial and archaeal genomes, i.e., around 10% of the total genomes contain a member of this family in low copy number.

## PemK-like, MazF-like (PFAM ID PF04014)

Toxin-antitoxin (TA) modules are fundamental for bacterial regulation upon environmental stresses. *mazEF* encodes the MazF toxin and its cognate MazE antitoxin. MazE possesses an N-terminal DBD through which it regulates its own promoter. The structure of the MazE-MazF complex is hexameric and is comprised of a MazE homodimer sandwiched between

MazF homodimers [38]. MazE binds into the major groove of double-stranded DNA *'a'*, involving side-chains of residues W9, N11, and R16 for the main interactions with the oligonucleotide. Moreover, the structural model resembles strongly the structure of *Rickettsia felis* VapB₂ (*Rf*VapB₂) in complex with its operator. While cataloged as "VapB" due to its association with a VapC toxin, this antitoxin contains an AbrB-type DBD similar to *Ec*MazE. Indeed, *Rf*VapB₂ recognizes the same palindrome as *Ec*MazE, using identical interactions with the N-terminal β-strand and hairpin. Interactions differ nevertheless at the periphery of the combining site, where the structures of both proteins diverge. There, alternative contacts are seen, such as the backbone NH of A19 in *Ec*MazE mimicking the interaction of the side chain of K19 from *R. felis* VapB with a DNA backbone phosphate [39]. This family is present in 45% of the total of the genomes, showing that they are relevant to the functioning of this system in a large diversity of organisms.

## Ribbon-helix-helix TFs (PF01402)

The ribbon-helix-helix (RHH) TFs bind to DNA through the insertion of an α-helix into the DNA major groove, a motif which is used by the ubiquitous H-T-H family of TFs; however, RHH proteins use an anti-parallel β-sheet to recognize specific nucleotide sequences and α-helices to anchor the β-sheet in the DNA major groove [40]. The distribution of this family (51% of the organisms analyzed contain at least one member) suggests an important role in the regulation of diverse but fundamental processes in the cell, such as methionine metabolism [40].

## Genomic context indicates conservation and diversity of regulated functions

Diverse analysis has shown that genomic context can be useful to predict the function of genes regulated by TFs [41]. This approach is based on the observation that the genes coding for the TFs and target genes are often transcribed in opposite directions from overlapping or adjacent promoters, as occurs in the LysR family [42]; i.e., the transcriptional activation of the target gene occurs when one dimer of the TF binds to a binding site located adjacent the -35 box of the target gene promoter and interacts with the RNA polymerase. At the same time, the self-repression of the TF occurs when it binds to a TF-binding site that overlaps its own promoter located on the opposite strand of the DNA [42]. In this regard, in order to dissect functionally the repertoire of predicted TFs and their relationship with neighbor genes, we evaluated the genome context for the 10 most abundant families identified in the collection of TFs and that represents the 60% of the total of TFs (Table 2). The analysis describes the orientation, length of intergenic region, and function of adjacent genes. To this end, the TFs were divided in two main groups: those TFs with a divergent orientation with the neighbor gene and those TFs likely to be cotranscribed with an upstream or downstream gene, as they are in the same orientation and they belong to the same TU. From Table 2, a high proportion of genes encoding TFs of the families related to the two-component system, such as OmpR/PhoB, LuxR/UhpA and Fis, are organized in TUs, mainly with those genes encoding sensor proteins. We must remember that the two-component system requires two elements, the sensor and response regulators, mainly cotranscribed. In addition, in LysR and TetR/AcrR families, there is a high proportion of divergently oriented genes relative to their neighbors. In this regard, for the genes divergently transcribed, most regulators are separated from their divergent partners by 100 bp or less (Fig 3), which suggests that the majority of the divergently oriented TF-related genes could regulate the adjacent gene.

**Table 2. Distributions of genes related to the studied protein families.**

| Family | PFAM ID | Total TFs | No. (%) of genes | | | |
|---|---|---|---|---|---|---|
| | | | Divergent | In TUs | Enzyme-associated | Virulence-associated |
| LuxR/UhpA | PF00196 | 12,782 | 2226 (0.17) | 3882 (30.37) | 313 (2.45) | 1980 (15.49) |
| OmpR/PhoB | PF00486 | 14,418 | 2404(0.16) | 7038 (48.81) | 362 (2.51) | 3130 (21.71) |
| Fis | PF02954 | 7103 | 1117 (0.15) | 2745 (38.65) | 991 (13.95) | 3311 (46.61) |
| LysR | PF00126 | 21,219 | 6380 (0.30) | 1747 (8.23) | 225 (1.06) | 1433 (6.75) |
| GalR/LacI | PF00356 | 7063 | 1555 (0.22) | 1299 (18.39) | 162 (2.29) | 911 (12.90) |
| GntR | PF00392 | 13,086 | 3038 (0.23) | 2989 (22.84) | 121 (0.92) | 1 (0.01) |
| TetR/ AcrR | PF00440 | 23,556 | 5965 (0.25) | 4273 (18.14) | 204 (0.87) | 57 (0.24) |
| MarR | PF01047 | 5841 | 1295 (0.22) | 1543 (26.42) | 41 (0.70) | 165 (2.82) |
| HTH_3 | PF01381 | 11,931 | 2408 (0.20) | 3441 (28.84) | 385 (3.23) | 576 (4.83) |
| HTH_18 | PF12833 | 15,987 | 3710 (0.23) | 2570 (16.08) | 3226 (20.18) | 719 (4.50) |

Columns are as follows: Family protein; PFAM ID; Total of proteins per family; % of divergent genes; % of genes in Transcription Units; Total and % of neighbor genes associated to enzymes and virulence. In brackets are indicated the % of proteins associated with each category.

## Functional analysis of divergently oriented adjacent genes

In order to evaluate the functions associated with groups of TFs and their divergent genes, diverse analyses were achieved, such as annotations based on Enzyme Commission (EC) numbers, the identification of proteins involved in virulence, and PFAM annotations. Therefore, based on CatFam assignments [23], two families exhibiting a high proportion of enzymatic activities, with 20% of the repertoire of HTH_18 and 13.95% of members of the Fis family, were also assigned as enzymes. This finding suggests that some proteins devoted to gene regulation are also associated with enzymatic activities, such as those proteins associated with the Fis family, classically described as Enhancer-binding proteins or EBP, whose functioning involves "higher-order" oligomer formation, and ATP hydrolysis coupled to the restructuring of the RNA polymerase [43, 44].

Therefore, a functional classification based on the EC numbers (according to CatFam) was achieved for both TFs and divergent neighboring protein-encoding genes. Based on this analysis, we found that the most abundant function is associated with transferases (EC number 2.-.-.-) (Table 2 and Fig 4), except for OmpR/PhoB, where around 80% of the TFs identified with an enzymatic activity correspond to lyases (EC number 4.-.-.-), which have been associated with the process of ATP catalysis. A similar distribution was found in neighbor genes where the EC number 2.-.-.-, or transferases, are also the most abundant. This finding correlates with a recent finding concerning in metabolism enzymes, where transferases are highly abundant, suggesting that enzyme-catalyzed transfer reactions are highly abundant in metabolism, probably because one of the most recurrent enzymatic activities identified in all the organisms corresponds to transferases of phosphorus-containing groups (2.7.-.-), in particular, the nucleotidyl phosphotransferases (2.7.7.-) involved in the transfer of acyl, glycosyl, amino, and phosphate (includes diphosphate, nucleotidyl residues, and others) [45]. Finally, the TFs of the OmpR/PhoB family also contain enzymes with the function of isomerases, probably because carbon source metabolism, where regulators of this family play a fundamental role, involves diverse isomerization processes [46]. S1 File.

## TFs and neighbor-related genes involved in virulence processes

In order to evaluate if TFs from particular families and neighbor genes are related to virulence processes, protein sequence comparison to identify orthologs from a virulence factor database

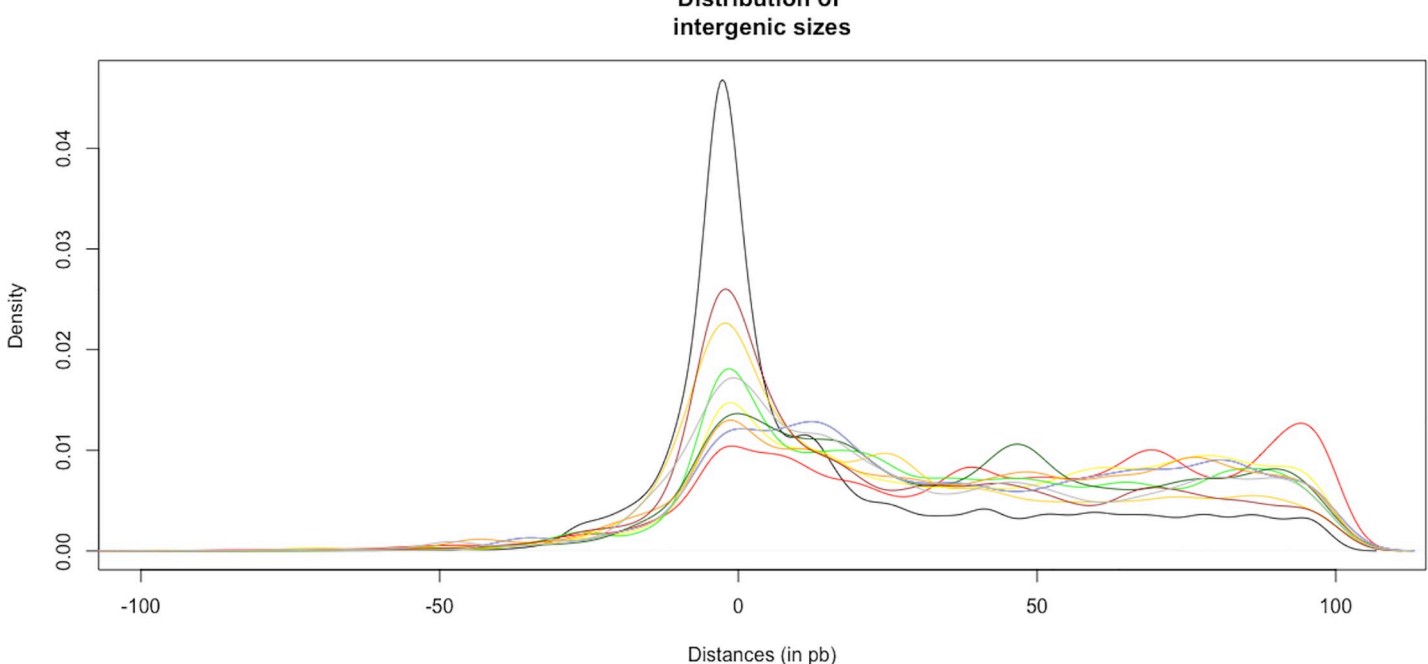

**Fig 3. Distributions of intergenic regions of the divergent genes and regulators.** The x axis shows length between the TFs and their divergent genes (in base pairs). Colors represent each TF family. PF00356, GalR/LacI (blue); PF00126, LysR (red); PF00440, TetR/ AcrR (yellow); PF00392, GntR (green); PF01047, MarR (darkgreen); PF12833, HTH_18 (orange); PF02954, Fis (gold); PF00196, LuxR/UhpA (brown); PF01381, HTH_3 (grey); PF00486, OmpR/PhoB(pink).

was achieved [20], as described in methods. From this analysis, it was found that three families contain a high proportion of virulence proteins, OmpR/PhoB, LuxR/UhpA, and Fis, which account for 21.71%, 15.49%, and 46.61%, respectively. Table 2 and Fig 5. In addition, the neighbor genes from OmpR/PhoB, Fis, and TetR/AcrR account for proteins associated with virulence, with 17.7%, 17.1%, and 11.5%, respectively. This finding is interesting because it indicates that TFs and neighbor genes for the OmpR and Fis families of TFs are involved in virulence. S2 File.

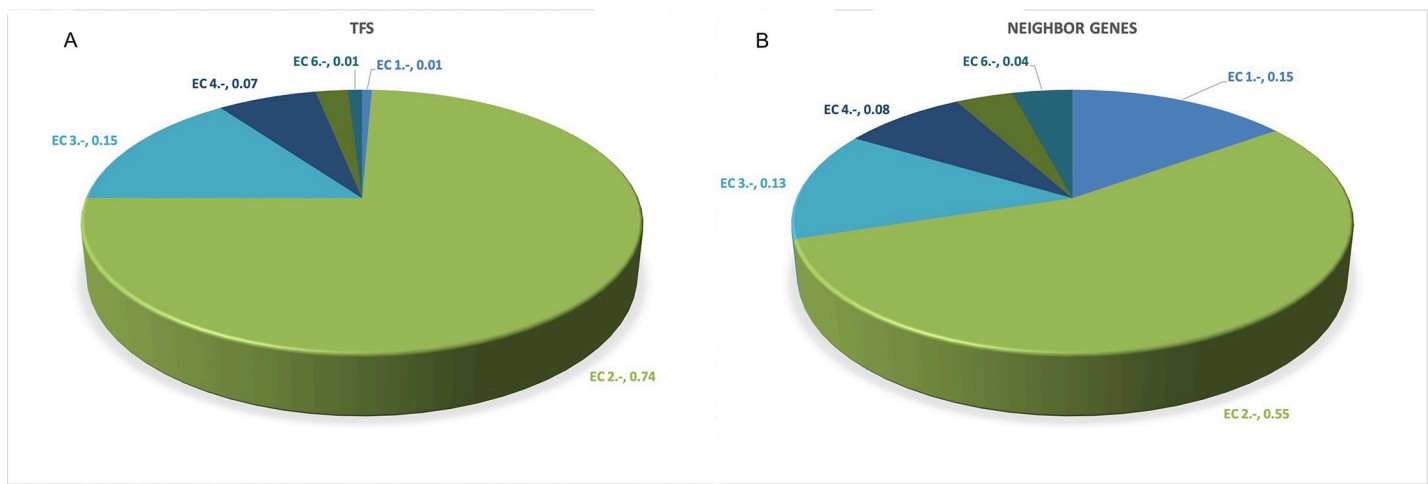

**Fig 4.** Enzymatic activities identified for A) TFs and for B) protein encoded by neighbor genes. The proportion of each of the six enzymatic numbers EC:1.- Oxidoreductases;. EC:2.- Transferases; EC:3.- Hydrolases; EC:4.- Lyases; EC:5.- Isomerases; EC:6.- Ligases, is indicated

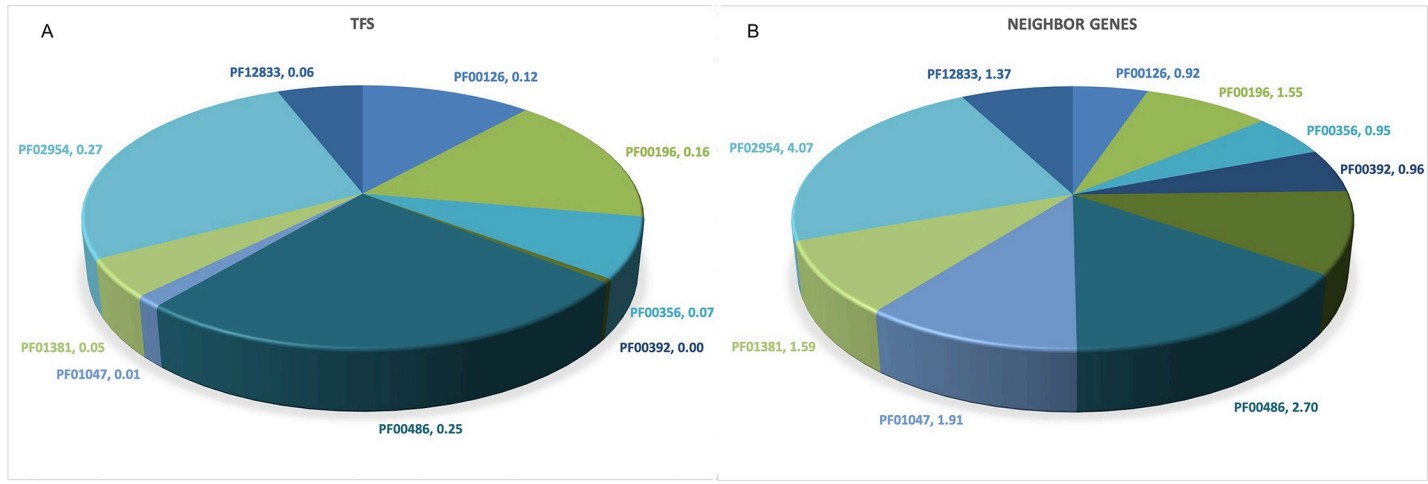

**Fig 5.** Virulence proteins associated with A) TFs and for B) proteins encoded by neighbor genes. The proportion of proteins associated to virulence is indicated. Families are as follow: PF00356, GalR/LacI PF00126, LysR; PF00440, TetR/ AcrR; PF00392, GntR; PF01047, MarR; PF12833, HTH_18 PF02954, Fis; PF00196, LuxR/ UhpA; PF01381, HTH_3; PF00486, OmpR/PhoB.

### Functional associations of TFs and neighboring genes

Finally, we evaluated the function of neighbor genes oriented divergently relative to the TF-encoding genes based on PFAM assignments (see methods). From this analysis, we found that PFAMs annotated as small-molecule binding, and amino acid transmembrane transporter activity are enriched in neighbor genes of the LysR family, whereas those families including members of the two-component system are enriched with PFAMs associated with protein autophosphorylation, phosphorus metabolic processes, and small-molecule metabolic processes. Table 3. This finding correlates with the fact that two-component regulators must be phosphorylated to regulate gene expression. Finally, members of the MarR family are bordered by genes associated with cellular aromatic compound metabolic processes, responses to drug, stress, and abiotic stimuli, reinforcing the notion that this family is involved in adaptation and resistance to drugs [47, 48].

### Conclusions

In this work we identified the repertoire of DNA-binding TFs in bacteria and archaea following two approaches. The first approach considers annotations based on assignments via 111 PFAM models associated with TFs, whereas the second approach considers the identification of orthologs based on a manually curated collection of TFs. Based on these approaches, two main results can be described. First, the collection of TFs considers DNA-binding structures, such as the classical HTH, and less studied domains used to contact specifically the DNA, such as the LytTR or the ArfGap/RecO-like zinc finger domains. Although these domains are present in lower proportions compared with the classical families TetR/AcrR or LysR, our finding reveals the opportunity to identify novel or specific DNA-binding proteins devoted to regulation of gene expression, based on bioinformatics tools or experimental approaches. Second, based on PFAM annotation and orthology, collection of proteins involved in gene regulation in Bacteria and Archaea increases by approximately 10% the prediction described by Sanchez et al. [49], suggesting that characterized TFs do not have an evident domain to be classified by PFAM and that TFs collected have alternative DBDs. In addition, we found two different trends of TFs per bacterial or archaeal divisions, i.e., the repertoire of TFs in Actinobacteria,

**Table 3. Main functions identified by PFAM annotations of neighbor genes.**

| PFAM | GO term | p-value | FDR |
|---|---|---|---|
| PF00126 | Small molecule binding | 1.21E-10 | 4.37E-09 |
| | Identical protein binding | 1.38E-10 | 4.37E-09 |
| | Ion binding | 1.57E-06 | 1.35E-05 |
| | Protein dimerization activity | 5.12E-04 | 1.13E-03 |
| | Fatty acid derivative binding | 3.65E-06 | 2.47E-05 |
| PF00196 | Cell communication | 2.02E-07 | 7.85E-06 |
| | Response to oxygen-containing compound | 6.74E-07 | 1.74E-05 |
| | Multiorganism process | 1.36E-06 | 2.64E-05 |
| | Biosynthetic process | 5.19E-06 | 8.06E-05 |
| | Response to drug | 6.15E-06 | 8.12E-05 |
| PF00356 | Whole membrane | 1.73E-03 | 6.12E-03 |
| | Microbody | 2.51E-09 | 1.13E-07 |
| | ATPase complex | 1.33E-05 | 1.20E-04 |
| | Symplast | 8.80E-05 | 5.65E-04 |
| | Protein-DNA complex | 1.77E-03 | 6.12E-03 |
| PF00392 | Transporter activity | 3.06E-08 | 3.10E-07 |
| | Identical protein binding | 6.25E-05 | 1.77E-04 |
| | Small-molecule binding | 7.78E-05 | 1.97E-04 |
| | Ion binding | 7.56E-04 | 1.53E-03 |
| | Oxidoreductase activity, acting on CH-OH group of donors | 1.10E-09 | 3.89E-08 |
| PF00440 | Mitochondrion | 1.86E-08 | 2.79E-07 |
| | Organelle membrane | 4.77E-05 | 3.58E-04 |
| | Whole membrane | 4.13E-04 | 1.86E-03 |
| | Microbody | 1.28E-11 | 5.75E-10 |
| | Mitochondrial matrix | 1.12E-08 | 2.51E-07 |
| PF00486 | Phosphorus metabolic process | 2.97E-04 | 3.50E-03 |
| | Response to abiotic stimulus | 5.98E-04 | 5.07E-03 |
| | Protein autophosphorylation | 6.14E-05 | 1.45E-03 |
| | Anion transport | 1.14E-04 | 1.97E-03 |
| | Organic acid transport | 2.12E-04 | 2.80E-03 |
| PF01047 | Biosynthetic process | 7.87E-10 | 5.74E-08 |
| | Cellular aromatic compound metabolic process | 2.87E-08 | 1.44E-06 |
| | Organic cyclic compound metabolic process | 4.98E-08 | 1.56E-06 |
| | Heterocycle metabolic process | 6.18E-08 | 1.69E-06 |
| | Cellular nitrogen compound metabolic process | 7.73E-08 | 1.88E-06 |
| PF01381 | Small-molecule binding | 1.54E-05 | 1.08E-04 |
| | Identical protein binding | 3.91E-05 | 1.90E-04 |
| | Ion binding | 2.17E-04 | 7.15E-04 |
| | Transporter activity | 1.71E-03 | 3.55E-03 |
| | Carbohydrate derivative binding | 2.96E-03 | 5.52E-03 |
| PF02954 | Small-molecule metabolic process | 8.92E-11 | 9.55E-09 |
| | Phosphorus metabolic process | 1.59E-10 | 1.14E-08 |
| | Biosynthetic process | 1.46E-06 | 4.47E-05 |
| | Catabolic process | 1.22E-05 | 1.25E-04 |
| | Heterocycle metabolic process | 1.46E-05 | 1.42E-04 |
| PF12833 | Small-molecule metabolic process | 8.83E-12 | 7.65E-10 |
| | Phosphorus metabolic process | 2.38E-07 | 2.29E-06 |

*(Continued)*

**Table 3.** (Continued)

| PFAM | GO term | p-value | FDR |
|---|---|---|---|
| | Catabolic process | 9.25E-07 | 6.87E-06 |
| | Response to drug | 1.65E-06 | 1.16E-05 |
| | Biosynthetic process | 4.01E-05 | 1.52E-04 |

Columns are as follow: PFAM id, Gene ontology term; P-value and FDR.

Firmicute, and Deltaproteobacteria from Bacteria and in Euryarchaeota from Archaea cellular divisions fitting a linear fit function, whereas the repertoire of TFs in Gamma-, Alpha-, and Betaproteobacteria, the CFB group, and Cyanobacteria fitting a power function.

Finally, the neighbor genes identified in the top 10 families of TFs can be associated with diverse and important functions, such as virulence, enzymatic functions, phosphorylation mechanisms, and antibiotic resistance. Therefore, further analyses are necessary to corroborate experimentally the uncharacterized proteins analyzed in this work.

## Supporting information

**S1 Fig. Abundance of TFs in bacteria and archaeal genomes as a function of genome size in taxonomical groups.** The abundance of TFs for each genome is shown on the y axis. Each dot corresponds to one genome. The model function (black line) and $R^2$ are indicated for power: A) Alpha proteobacteria; B) Beta proteobacteria; C) Gamma proteobacteria; D) Cyanobacteria, and E) CFB (*Cytophaga*, *Fusobacterium*, and *Bacteroides*) group; and linear: F) Actinobacteria; G) Delta proteobacteria; H) Euryarchaeota; and I) Firmicutes.
(PDF)

**S1 File. Collection of TFs with enzymatic activities predicted with Catfam in nonredundant genomes (in text format).** Two columns are indicated per genome: Protein_ID and EC number.
(ZIP)

**S2 File. Collection of TFs with virulence functional role identified with VFDB in nonredundant genomes (in text format).** For each genome, six columns are indicated: Protein_ID derived from VFDB; Protein_ID (genome); e-value, bit score; e-value and bit score.
(ZIP)

## Acknowledgments

We thank to Joaquin Morales, Sandra Sauza and Israel Sanchez for their technical support.

## Author Contributions

**Conceptualization:** Ernesto Perez-Rueda.

**Data curation:** Alejandro Huerta-Saquero, Silvia Tenorio-Salgado, Nancy Rivera-Gomez, Alba Romero, Jose Antonio Ibarra, Ernesto Perez-Rueda.

**Formal analysis:** Emanuel Flores-Bautista, Rafael Hernandez-Guerrero, Ernesto Perez-Rueda.

**Resources:** Rafael Hernandez-Guerrero.

**Supervision:** Ernesto Perez-Rueda.

**Visualization:** Emanuel Flores-Bautista.

**Writing – original draft:** Alejandro Huerta-Saquero, Silvia Tenorio-Salgado, Nancy Rivera-Gomez, Jose Antonio Ibarra.

**Writing – review & editing:** Jose Antonio Ibarra, Ernesto Perez-Rueda.

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
