## [Decision Letter · Decision Letter 0]

10 Jun 2020

PONE-D-20-13420

Deciphering the functional diversity of DNA-binding transcription factors in Bacteria and Archaea organisms

PLOS ONE

Dear Dr. Ernesto Perez-Rueda, 

Thank you for submitting your manuscript to PLOS ONE. After careful consideration, we feel that it has merit but does not fully meet PLOS ONE’s publication criteria as it currently stands. Therefore, we invite you to submit a revised version of the manuscript that addresses the points raised during the review process.

We look forward to receiving your revised manuscript.

Kind regards,

Akira Ishihama, Ph.D.

Academic Editor

PLOS ONE

Journal Requirements:

Additional Editor Comments (if provided):

The authors intend to decipher the functional diversity of DNA-binding transcription factors from a total of 30 different species from bacteria and Archaea. I admire this laborious work, but at present, the regulatory functions of most TFs in a single species are known only for a limited number of organism such as Escherichia coli. this type approach is too early – I think. Nevertheless the report provides an idea for a front research in near future. Since both reviewers support publication only after substantial modifications and revisions, I will reconsider publication after checking the revised version. In the revised form, provide all figures in more readable high-quality formats and the supplemental data in reader-friendly formats.

Reviewers' comments:

Reviewer's Responses to Questions

**Comments to the Author**

1. Is the manuscript technically sound, and do the data support the conclusions?

Reviewer #1: Yes

Reviewer #2: Yes

2. Has the statistical analysis been performed appropriately and rigorously? 

Reviewer #1: N/A

Reviewer #2: Yes

3. Have the authors made all data underlying the findings in their manuscript fully available?

Reviewer #1: No

Reviewer #2: Yes

4. Is the manuscript presented in an intelligible fashion and written in standard English?

Reviewer #1: No

Reviewer #2: No

5. Review Comments to the Author

Reviewer #1: This is an important work that provides a valuable resource for studies of transcription factors. The effort that the authors put into curating their initial dataset is particularly impressive. However, there are significant issues in three areas that should be addressed in revision:

A: Additional clarifications are needed for the methods. Justification for selecting particular methods as opposed to available alternatives and selection of specific parameters should be provided.

B: The organization of the manuscript is appropriate and easy to follow. However, there are numerous occasions when the authors do not use the most accurate word or phrase. A few examples are provided in the specific comments below but this is a widespread issue throughout the manuscript. In some instances, the authors use language, possibly inadvertently, which appears to claim credit for discovering something that is already known. See specific comments below.

C: I was disappointed when I found that accessing the online database required a username and password. For that reason, I did not review the database and my comments relate only to the manuscript. I hope the authors intend to make the database publicly accessible without requiring the users to set up an account. I also did not review the supplementary data because they appear to require specialized software and there are no instructions how to use the files.

Specific comments:

1. In the Abstract, line 40, replace “identification” with “prediction.” This is still only a prediction, not a reliable determination of the TF’s function.

2. Line 54, “genomic organization resulting in contemporary systems” – consider rephrasing. I do not know what that means.

3. Line 61: “…express different genes in response to metabolic stimuli” – not only metabolic. For example, heat shock leads to changes in gene expression, or radiation damage, and many other non-metabolic stimuli.

4. Analogous to an earlier comment, on line 37, I would recommend using “… putatively devoted to gene regulation” unless these are experimentally verified results.

5. Some examples of awkward wording: On line 82, replace “have” with “prepare,” “create,” “design,” “obtain,” or some other appropriate word. On line 87, replace “checked and read” with “verified.” I am not sure what “including 842 references” in the same sentence refers to; it can probably be omitted. On line 89, replace “achieved” with “performed” or “used.” Rephrase “proteins with function beyond gene regulation” on line 91 – do you mean proteins that have other non-regulatory functions in addition to regulatory functions or proteins that do not have known regulatory functions? On line 93, “alternatively” should probably be replaced with “in addition.” Leave out “as a first approach” later in the same sentence.

6. I respect the authors’ choice to exclude sigma factor but I am mildly disappointed that they did not include them in this work. They have many similarities to transcription factors, in particular, both are DNA-binding regulatory proteins.

7. Can the authors provide some justification for choosing E<10^-5 when searching for orthologs? Did they test other values and select this cutoff after evaluating the results? How big effect does changing this parameter have on the results?

8. Is the method the authors used for operon predictions more accurate than alternative methods? For example, the method designed by Arkin lab (https://www.ncbi.nlm.nih.gov/pmc/articles/PMC549399/) uses comparative genomics in addition to distances between genes, which I would expect to provide a better accuracy. How robust are the results presented in this work relative to errors in the operon predictions? I assume that such errors probably do not matter very much but some discussion or data to support this assertion might be included in the paper because predicting operons is a difficult problem and I am not aware of any method that does it accurately.

9. Line 227: When referring to the supplementary figures, the authors should indicate specifically which figures. See also comments below on Supplementary material. I believe it should be presented in a more appropriate way. It might be also worthwhile to include some of the figures demonstrating this result directly in the manuscript.

10. Line 305: I would recommend changing “are transcribed in opposite directions” to “are often transcribed in opposite directions from overlapping or adjacent promoters.”

11. Line 331: What is “a significant distribution?”

12. Line 334: Change “proteins” to “some proteins.” I’d urge the authors to be more careful about similar over-generalizations that are not necessarily supported by their results. This is a widespread issue in the manuscript and I may not have noted all such instances in this review.

13. Line 356: Leave out “are also” from the title.

14. Line 366: “account” probably does not reflect the intended meaning of this sentence.

15. Line 369: this is another example when something is misleadingly presented as if it was a new discovery. pFAM was designed to improve functional annotations, so pointing out that it does exactly that after it had been widely used for that purpose for more than two decades seems unnecessary.

16. The statement on lines 386-389 again gives an impression that this is the first time someone found DNA-binding motifs other than HTH, which is not accurate.

17. The sentence on lines 396-401 needs to be revised for clarity. Do TFs actually grow?

18. Line 573: The authors should also clarify what they mean by “power distribution.” I initially thought they referred to power law but that does not seem to be the case. The relationship in the plot actually appears to be close to linear.

19. Figures are presented in such a poor quality that some labels are not legible. For example, I cannot make out the families in Figure 1.

20. Supplementary figures: I would strongly urge the authors to convert the supplementary figures into a PDF file and include appropriate legends, titles, and/or descriptions for all figures.

21. Other supplementary files include data that appear to require specialized software to view. I have not reviewed these files. There is no description or advice how these files can be accessed. I could not quickly find PLOS ONE policy on Supplementary Data to verify how the journal recommends handling such situations. If possible, I would suggest providing the data in some easily accessible format, such as tab-delimited text or Excel file. If that is not possible, detailed instructions on how to explore these data should be provided.

Reviewer #2: The manuscript presents an amazing resource

However the presentation is not very user-friendly. The Results section is mainly concerned with discussing exceptions and trends. Some of the Tables and Figures are poorly annotated with deficient legends. It would be more useful, first, to present the results from some bacteria, maybe a well known one versus a lesser known one? This would guide the Reader through the dataset. I was particularly interested to know how many of the predicted TFs really were complete TFs.

6. PLOS authors have the option to publish the peer review history of their article (what does this mean?). If published, this will include your full peer review and any attached files.

Reviewer #1: No

Reviewer #2: No

---

## [Author Response · Author response to Decision Letter 0]

30 Jun 2020

PONE-D-20-13420

Deciphering the functional diversity of DNA-binding transcription factors in Bacteria and Archaea organisms

PLOS ONE

Dear Dr. Ernesto Perez-Rueda, 

Thank you for submitting your manuscript to PLOS ONE. After careful consideration, we feel that it has merit but does not fully meet PLOS ONE’s publication criteria as it currently stands. Therefore, we invite you to submit a revised version of the manuscript that addresses the points raised during the review process.

● A rebuttal letter that responds to each point raised by the academic editor and reviewer(s). You should upload this letter as a separate file labeled 'Response to Reviewers'.

● A marked-up copy of your manuscript that highlights changes made to the original version. You should upload this as a separate file labeled 'Revised Manuscript with Track Changes'.

● An unmarked version of your revised paper without tracked changes. You should upload this as a separate file labeled 'Manuscript'.

We look forward to receiving your revised manuscript.

Kind regards,

Akira Ishihama, Ph.D.

Academic Editor

PLOS ONE

Journal Requirements:

Additional Editor Comments (if provided):

The authors intend to decipher the functional diversity of DNA-binding transcription factors from a total of 30 different species from bacteria and Archaea. I admire this laborious work, but at present, the regulatory functions of most TFs in a single species are known only for a limited number of organism such as Escherichia coli. this type approach is too early – I think. Nevertheless the report provides an idea for a front research in near future. Since both reviewers support publication only after substantial modifications and revisions, I will reconsider publication after checking the revised version. In the revised form, provide all figures in more readable high-quality formats and the supplemental data in reader-friendly formats.

RESPONSE: Thank you very much for your comment. With the changes suggested by the reviewers and the improved presentation of figures and supplemental data, we are sure that our work fits requirements to be published in this prestigious journal.

Reviewers' comments:

Reviewer's Responses to Questions

Comments to the Author

1. Is the manuscript technically sound, and do the data support the conclusions?

Reviewer #1: Yes

Reviewer #2: Yes

2. Has the statistical analysis been performed appropriately and rigorously? 

Reviewer #1: N/A

Reviewer #2: Yes

3. Have the authors made all data underlying the findings in their manuscript fully available?

Reviewer #1: No

Reviewer #2: Yes

4. Is the manuscript presented in an intelligible fashion and written in standard English?

Reviewer #1: No

Reviewer #2: No

5. Review Comments to the Author

Reviewer #1: This is an important work that provides a valuable resource for studies of transcription factors. The effort that the authors put into curating their initial dataset is particularly impressive. However, there are significant issues in three areas that should be addressed in revision:

A: Additional clarifications are needed for the methods. Justification for selecting particular methods as opposed to available alternatives and selection of specific parameters should be provided.

RESPONSE: We have improved the methods to clarify their implementation in this work. For instance, we included diverse justifications along the method, as described for the orthologous analysis:

To identify orthologous proteins, the program Proteinortho [18] was applied. We used this program because it implements an extended version of the reciprocal best heuristic alignment [18]; reduces the amount of memory required for orthology analysis, when compared to OrthoMCL and Multi-Paranoid, and the performance is comparable with OrthoMCL[19]. To this end, we computed the orthologous proteins in the complete set of all 668 proteins against the 5321 bacterial and archaeal genomes, employing an E-value ≤ 10-5, with a coverage of ≥70%, considered as significant to identify orthologous proteins against the collection of well-known proteins.

Concerning the operon predictions, we included this paragraph:

This method was used because of its high performance to identify correctly operons for E. coli and B. subtilis [22]. In addition, divergently oriented genes relative to the TF and their intergenic distances were computed.

and Catfam…

To determine if TFs and their neighbor genes are associated with enzymatic activities, the Catalytic Families (CatFam version 2.0) program was used to scan the complete set of proteins associated with the 1321 representative bacterial and archaeal genomes, using default conditions. We used, CatFam because it generates 8880 sequence profiles through a stepwise procedure that carefully controls profile quality and employs nonenzymes as negative samples to establish profile-specific thresholds associated with a predefined nominal false-positive rate of predictions; i.e. it predicts enzymes with 98.6% precision and 95.0% recall [23].

B: The organization of the manuscript is appropriate and easy to follow. However, there are numerous occasions when the authors do not use the most accurate word or phrase. A few examples are provided in the specific comments below but this is a widespread issue throughout the manuscript. In some instances, the authors use language, possibly inadvertently, which appears to claim credit for discovering something that is already known. See specific comments below.

RESPONSE: We have corrected and clarified all these expressions along the manuscript.

C: I was disappointed when I found that accessing the online database required a username and password. For that reason, I did not review the database and my comments relate only to the manuscript. I hope the authors intend to make the database publicly accessible without requiring the users to set up an account. I also did not review the supplementary data because they appear to require specialized software and there are no instructions how to use the files.

RESPONSE: In the first submitted version we have included a temporal “login and password” to the reviewers; perhaps these expired during this review and for that we apologize. For the revised version we have made this database accessible without requiring the users to set up an account, at http://web.pcyt.unam.mx/EntrafDB/

Specific comments:

1. In the Abstract, line 40, replace “identification” with “prediction.” This is still only a prediction, not a reliable determination of the TF’s function.

RESPONSE: We have replaced this expression.

2. Line 54, “genomic organization resulting in contemporary systems” – consider rephrasing. I do not know what that means.

RESPONSE:We have clarified this sentence. We refer to:

...the genomes are the products of diverse evolutionary events, such as gene expansion, gene loss, and lateral gene transfer.

3. Line 61: “…express different genes in response to metabolic stimuli” – not only metabolic. For example, heat shock leads to changes in gene expression, or radiation damage, and many other non-metabolic stimuli.

RESPONSE: We have modified the sentence:

“express different genes in response to internal and external stimuli“

4. Analogous to an earlier comment, on line 37, I would recommend using “… putatively devoted to gene regulation” unless these are experimentally verified results.

RESPONSE: We have modified the sentence.

5. Some examples of awkward wording: On line 82, replace “have” with “prepare,” “create,” “design,” “obtain,” or some other appropriate word. On line 87, replace “checked and read” with “verified.” I am not sure what “including 842 references” in the same sentence refers to; it can probably be omitted. On line 89, replace “achieved” with “performed” or “used.” Rephrase “proteins with function beyond gene regulation” on line 91 – do you mean proteins that have other non-regulatory functions in addition to regulatory functions or proteins that do not have known regulatory functions? On line 93, “alternatively” should probably be replaced with “in addition.” Leave out “as a first approach” later in the same sentence.

RESPONSE: We have corrected all these expressions.

6. I respect the authors’ choice to exclude sigma factor but I am mildly disappointed that they did not include them in this work. They have many similarities to transcription factors, in particular, both are DNA-binding regulatory proteins.

RESPONSE: We agree with the reviewer’s comment regarding sigma factors as DNA binding proteins, but the main reason to exclude them in this work because we consider the follow definition for a Transcriptional factor:

To this end, we defined a TF as DNA-binding protein needed to activate or repress the transcription of a gene, but are themselves neither part of the RNA polymerase (RNAP) core nor of the holoenzyme [24]. Therefore, sigma factors were not considered as TFs in this study.

7. Can the authors provide some justification for choosing E<10^-5 when searching for orthologs? Did they test other values and select this cutoff after evaluating the results? How big effect does changing this parameter have on the results?

RESPONSE: The identification of orthologs was complementary to the HMM searches. In this regard, two coverages were considered, the first one, by 100% and the second one, 70%. We considered that 70% of coverage, an e-value of less than 10-5 and a bidirectional best hit is significant to identify orthologous proteins against the collection of well-known proteins. We added a paragraph describing this information.

To evaluate the accuracy of the prediction process, we compared our predictions against the repertoire described in bacterial models. For instance, in E. coli K12 we predicted a total of 336 TFs, from these, 103 proteins were exclusively predicted by HMM searches, 203 by PFAM models and by sequence comparisons against the well-known dataset, whereas 30 proteins were only predicted by sequence comparisons. In B. subtilis 168, we predicted 286 TFs, 118 by HMM searches, 123 by both approaches, and 43 by sequence comparisons. Thus, 8.9% and 15.0% of the predictions of TFs in E. coli and B. subtilis, respectively, were only obtained by sequence comparisons, suggesting that all predictions are increasing when our dataset of experimentally described TFs is considered. 

8. Is the method the authors used for operon predictions more accurate than alternative methods? For example, the method designed by Arkin lab (https://www.ncbi.nlm.nih.gov/pmc/articles/PMC549399/) uses comparative genomics in addition to distances between genes, which I would expect to provide a better accuracy. How robust are the results presented in this work relative to errors in the operon predictions? I assume that such errors probably do not matter very much but some discussion or data to support this assertion might be included in the paper because predicting operons is a difficult problem and I am not aware of any method that does it accurately.

RESPONSE: We consider that the approach described by Moreno-Hagelsieb and Collado-Vides (2002) is robust to predict operons. In this regard, a comparative analysis described by Brouwer et al., (2008) described that the predictions used in this work contain a high rate of true positives and true negatives, probably because they used a larger number of verified operons available; i.e. high performance for E. coli and B. subtilis. In contrast, the performance of the method by Price et al., is much better for E. coli than for B. subtilis; however, as the reviewer comments, the operon prediction is an open topic that does not affect the central analysis described in this manuscript.

Rutger W. W. Brouwer, Oscar P. Kuipers, Sacha A. F. T. van Hijum. (2008). The relative value of operon predictions. Briefings in Bioinformatics, 9(5):367–375, https://doi.org/10.1093/bib/bbn019

9. Line 227: When referring to the supplementary figures, the authors should indicate specifically which figures. See also comments below on Supplementary material. I believe it should be presented in a more appropriate way. It might be also worthwhile to include some of the figures demonstrating this result directly in the manuscript.

RESPONSE: We have cited appropriately the supplementary material in the manuscript.

10. Line 305: I would recommend changing “are transcribed in opposite directions” to “are often transcribed in opposite directions from overlapping or adjacent promoters.”

RESPONSE: We have changed the sentence.

11. Line 331: What is “a significant distribution?”

RESPONSE: We refer to “high proportion”. It has been changed in the manuscript.

12. Line 334: Change “proteins” to “some proteins.” I’d urge the authors to be more careful about similar over-generalizations that are not necessarily supported by their results. This is a widespread issue in the manuscript and I may not have noted all such instances in this review.

RESPONSE: We have modified all these generalizations.

13. Line 356: Leave out “are also” from the title.

RESPONSE: We have modified the title.

14. Line 366: “account” probably does not reflect the intended meaning of this sentence.

RESPONSE: We modified the sentence:

“This finding is interesting because it indicates that TFs and neighbor genes for the OmpR and Fis families of TFs are involved in virulence”

15. Line 369: this is another example when something is misleadingly presented as if it was a new discovery. pFAM was designed to improve functional annotations, so pointing out that it does exactly that after it had been widely used for that purpose for more than two decades seems unnecessary.

RESPONSE: We have modified the title, excluding the obvious:

“Functional associations of TFs and neighboring genes”

16. The statement on lines 386-389 again gives an impression that this is the first time someone found DNA-binding motifs other than HTH, which is not accurate.

RESPONSE:We have modified the sentence:

“First, the collection of TFs considers DNA-binding structures, such as the classical HTH, and less studied domains used to contact specifically the DNA…” 

17. The sentence on lines 396-401 needs to be revised for clarity. Do TFs actually grow?

RESPONSE: We have modified this expression, describing the distribution of TFs. In this regard, we found a positive correlation between genome size and number of TFs.

18. Line 573: The authors should also clarify what they mean by “power distribution.” I initially thought they referred to power law but that does not seem to be the case. The relationship in the plot actually appears to be close to linear.

RESPONSE: The TFs follow a power-law distribution. We have added the follow paragraph:

Based on this approach, the Pearson correlation coefficient was 0.88 (p-value < 2.2e216), showing a strong positive correlation between TFs and genome size (measured by ORFs number). In addition, the power-law fitting function exponent (1.52) was within the range reported in other studies for protein families classified as regulators [30].

19. Figures are presented in such a poor quality that some labels are not legible. For example, I cannot make out the families in Figure 1.

RESPONSE: We have improved all the figures.

20. Supplementary figures: I would strongly urge the authors to convert the supplementary figures into a PDF file and include appropriate legends, titles, and/or descriptions for all figures.

RESPONSE: We have improved all the supplementary material.

21. Other supplementary files include data that appear to require specialized software to view. I have not reviewed these files. There is no description or advice how these files can be accessed. I could not quickly find PLOS ONE policy on Supplementary Data to verify how the journal recommends handling such situations. If possible, I would suggest providing the data in some easily accessible format, such as tab-delimited text or Excel file. If that is not possible, detailed instructions on how to explore these data should be provided.

RESPONSE: We have improved all the supplementary material and a website describing all predictions per genome is available at:

http://web.pcyt.unam.mx/EntrafDB/

 

Reviewer #2: The manuscript presents an amazing resource

However the presentation is not very user-friendly. The Results section is mainly concerned with discussing exceptions and trends. Some of the Tables and Figures are poorly annotated with deficient legends. It would be more useful, first, to present the results from some bacteria, maybe a well known one versus a lesser known one? This would guide the Reader through the dataset. I was particularly interested to know how many of the predicted TFs really were complete TFs.

RESPONSE: We have improved the manuscript, including diverse paragraphs, such as the following:

Finally, the experimentally characterized TFs are mainly associated with repression of genes (29.4%), followed by TFs that can activate and/or repress gene expression, 23.9%, and a low proportion of activators (18.1%). Proteins with functions not clearly defined represents the 28% of the collection. This finding suggests that repression is the most abundant regulatory mechanisms, as it has been described in the E. coli K12 [10] and B. subtilis 168 [4], and it is consistent with hypothesis that most promoters are repressed in bacteria, mainly by steric hindrance, where the repressor-binding site overlaps core promoter elements and blocks recognition of the promoter by the RNAP holoenzyme [29].

Or the next one.

We consider that our set of known and predicted TFs in both genomes is similar to previous works [32] [4, 33], and that we are close to the total number of TFs in these bacteria. Therefore, the existence of alternative regulatory mechanisms, such as riboswitches, DNA-curvature or attenuation could influence gene expression where there is no evidence of regulation mediated by TFs, as occurs in organisms where less proportion of TFs was predicted; such as occurs in the delta proteobacterium Sorangium cellulosum genome (GCF_000418325) with 10514 ORFs and 402 predicted TFs, i.e. 3.8% of its gene products. In particular, Hang et al [34] identified a significant abundance of eukaryote-like protein kinases (508 proteins), 169 sigma factors, 6 anti-sigma factor proteins, and 347 sigma factor-related proteins; suggesting that the regulation of gene products are mainly related to kinases and sigma factors than TFs [34]. 

6. PLOS authors have the option to publish the peer review history of their article (what does this mean?). If published, this will include your full peer review and any attached files.

Do you want your identity to be public for this peer review? For information about this choice, including consent withdrawal, please see our Privacy Policy.

Reviewer #1: No

Reviewer #2: No

---

## [Decision Letter · Decision Letter 1]

15 Jul 2020

PONE-D-20-13420R1

Deciphering the functional diversity of DNA-binding transcription factors in Bacteria and Archaea organisms

PLOS ONE

Dear Dr. Ernesto Perez-Rueda,

Thank you for submitting your manuscript to PLOS ONE. After careful consideration, we feel that it has merit but does not fully meet PLOS ONE’s publication criteria as it currently stands. Therefore, we invite you to submit a revised version of the manuscript that addresses the points raised during the review process.

We look forward to receiving your revised manuscript.

Kind regards,

Akira Ishihama, Ph.D.

Academic Editor

PLOS ONE

Reviewers' comments:

Reviewer's Responses to Questions

**Comments to the Author**

1. If the authors have adequately addressed your comments raised in a previous round of review and you feel that this manuscript is now acceptable for publication, you may indicate that here to bypass the “Comments to the Author” section, enter your conflict of interest statement in the “Confidential to Editor” section, and submit your "Accept" recommendation.

Reviewer #1: (No Response)

2. Is the manuscript technically sound, and do the data support the conclusions?

Reviewer #1: Yes

3. Has the statistical analysis been performed appropriately and rigorously? 

Reviewer #1: Yes

4. Have the authors made all data underlying the findings in their manuscript fully available?

Reviewer #1: Yes

5. Is the manuscript presented in an intelligible fashion and written in standard English?

Reviewer #1: Yes

6. Review Comments to the Author

Reviewer #1: The authors addressed adequately most of my comments. One important issue that has not been resolved is the confusing use of the term “distribution” when the authors refer to a relationship between two variables. For example, the plots in supplementary figures do not show power law or linear distributions but relationships between pairs of variables that appear to fit a power function or a linear function. Also note that in a power-law probability distribution, the exponent in the power function has to be negative. I believe this should be corrected before publication.

In some instances, the authors use the word “lineal” when they probably mean “linear.”

As a less important issue, I would still urge the authors to consider converting the bitmap files in the zip archive S1 into a PDF file where the figures could be presented with appropriate explanatory legends.

7. PLOS authors have the option to publish the peer review history of their article (what does this mean?). If published, this will include your full peer review and any attached files.

Reviewer #1: No

---

## [Author Response · Author response to Decision Letter 1]

17 Jul 2020

Reviewer's Responses to Questions

Comments to the Author

1. If the authors have adequately addressed your comments raised in a previous round of review and you feel that this manuscript is now acceptable for publication, you may indicate that here to bypass the “Comments to the Author” section, enter your conflict of interest statement in the “Confidential to Editor” section, and submit your "Accept" recommendation.

Reviewer #1: (No Response)

2. Is the manuscript technically sound, and do the data support the conclusions?

Reviewer #1: Yes

3. Has the statistical analysis been performed appropriately and rigorously?

Reviewer #1: Yes

4. Have the authors made all data underlying the findings in their manuscript fully available?

Reviewer #1: Yes

5. Is the manuscript presented in an intelligible fashion and written in standard English?

Reviewer #1: Yes

6. Review Comments to the Author

Reviewer #1: The authors addressed adequately most of my comments. One important issue that has not been resolved is the confusing use of the term “distribution” when the authors refer to a relationship between two variables. For example, the plots in supplementary figures do not show power law or linear distributions but relationships between pairs of variables that appear to fit a power function or a linear function. Also note that in a power-law probability distribution, the exponent in the power function has to be negative. I believe this should be corrected before publication.

RESPONSE: We appreciate this comment. In effect, we have corrected the term “distribution” in the manuscript. This relationship refers to the association between the total of TFs against the genome size, and the “power or linear” make reference to the model function fit associated to this association.

In some instances, the authors use the word “lineal” when they probably mean “linear.”

RESPONSE: We have corrected this word. It refers to “linear fit function”.

As a less important issue, I would still urge the authors to consider converting the bitmap files in the zip archive S1 into a PDF file where the figures could be presented with appropriate explanatory legends.

RESPONSE: We have converted the supplementary figures of archive S1 into a PDF file, and the appropriate footnote was included in the manuscript.

7. PLOS authors have the option to publish the peer review history of their article (what does this mean?). If published, this will include your full peer review and any attached files.

Do you want your identity to be public for this peer review? For information about this choice, including consent withdrawal, please see our Privacy Policy.

Reviewer #1: No

---

## [Editor Report · Decision Letter 2]

22 Jul 2020

Deciphering the functional diversity of DNA-binding transcription factors in Bacteria and Archaea organisms

PONE-D-20-13420R2

Dear Dr. Ernesto Perez-Rueda,

We’re pleased to inform you that your manuscript has been judged scientifically suitable for publication and will be formally accepted for publication once it meets all outstanding technical requirements.

Kind regards,

Akira Ishihama, Ph.D.

Academic Editor

PLOS ONE
---

## [Editor Report · Acceptance letter]

12 Aug 2020

PONE-D-20-13420R2 

Deciphering the functional diversity of DNA-binding transcription factors in Bacteria and Archaea organisms 

Dear Dr. Perez-Rueda:

I'm pleased to inform you that your manuscript has been deemed suitable for publication in PLOS ONE. Congratulations! Your manuscript is now with our production department. 

Kind regards, 

on behalf of

Professor Akira Ishihama 

Academic Editor

PLOS ONE